Subject Areas:
computational mechanics/materials science/civil engineering

Keywords:
self-shaping, hygromorphs, wood bilayer, gridshell, Gaussian curvature

Author for correspondence:
Philippe Grönquist
e-mail: groenquist@ibk.baug.ethz.ch

# Computational analysis of hygromorphic self-shaping wood gridshell structures

Philippe Grönquist[1,2,3], Prijanthy Panchadcharam[2], Dylan Wood[4], Achim Menges[4], Markus Rüggeberg[1,2] and Falk K. Wittel[2]

[1]Laboratory for Cellulose & Wood Materials, Empa, 8600 Dübendorf, Switzerland
[2]Institute for Building Materials, ETH Zurich, 8093 Zürich, Switzerland
[3]Institute of Structural Engineering, ETH Zurich, 8093 Zürich, Switzerland
[4]Institute for Computational Design and Construction, University of Stuttgart, 70174 Stuttgart, Germany

PG, 0000-0001-8432-1706; DW, 0000-0003-0922-5399;
AM, 0000-0001-9055-4039; MR, 0000-0002-6966-8311;
FKW, 0000-0001-8672-5464

Bi-layered composites capable of self-shaping are of increasing relevance to science and engineering. They can be made out of anisotropic materials that are responsive to changes in a state variable, e.g. wood, which swells and shrinks by changes in moisture. When extensive bending is desired, such bilayers are usually designed as cross-ply structures. However, the nature of cross-ply laminates tends to prevent changes of the Gaussian curvature so that a plate-like geometry of the composite will be partly restricted from shaping. Therefore, an effective approach for maximizing bending is to keep the composite in a narrow strip configuration so that Gaussian curvature can remain constant during shaping. This represents a fundamental limitation for many applications where self-shaped double-curved structures could be beneficial, e.g. in timber architecture. In this study, we propose to achieve double-curvature by gridshell configurations of narrow self-shaping wood bilayer strips. Using numerical mechanical simulations, we investigate a parametric phase-space of shaping. Our results show that double curvature can be achieved and that the change in Gaussian curvature is dependent on the system's geometry. Furthermore, we discuss a novel architectural application potential in the form of self-erecting timber gridshells.

## 1. Introduction

Programmable self-shaping materials are widely studied, e.g. at small scale in fields such as soft (or hard) actuated matter

**Figure 1.** Curvatures of a bilayer-composite. (*a*) Two elastic sheets of thickness $h/2$ are stretched by a factor of $\lambda$ and then bonded together. The same configuration is achievable by first bonding two stimuli-responsive layers together and inducing $\lambda$ with the stimuli, e.g. a moisture-content change in wood will result in anisotropic swelling- or shrinkage-induced stretches in the single layers. (*b*) Upon release, the bilayer-composite plate-like sheet self-shapes into an exemplary saddle configuration with principal curvatures $\kappa_1$ and $\kappa_2$. (*c*) A narrow quasi-two-dimensional strip cut out from the sheet displays natural curvature $\kappa_0$ and stretch $\lambda_0$ (on the basis of [20,21]).

[1–9], or at medium and large scale mostly for adaptive structures in construction applications [10–17]. A commonly known principle of obtaining self-shaping systems is the bilayer-laminate. A residual stress state, responsible for the shaping, can be obtained by two bonded materials with different expansion or retraction properties with respect to a controllable state-variable such as the moisture content [18,19]. In the case of narrow bilayer strips, the resulting differential strain ($\varepsilon_0$, or stretch $\lambda_0 = \varepsilon_0 + 1$) across the thickness ($h$) direction is directly proportional to the induced curvature of shaping ($\kappa_0 = h^{-1}\varepsilon_0$). This type of curvature is often termed *natural curvature* [20,21] since in quasi-two-dimensional configuration, bilayer strips are free from shaping restrictions of plate-like geometries such as shown in figure 1. For plate geometries, the shaping can be very complex especially for anisotropic composites, e.g. a laminate of two plies of the same material with different fibre or strong direction with respect to each other. Often, a cross-ply structure is chosen, which is maximizing the amount of differential strain. However, such configurations tend to exclusively achieve saddle-like shapes, similar to the example in figure 1*b*. The reason is that cross-ply structures of stiff anisotropic composites prevent significant in-plane stretching, which results in very limited and negative changes in *Gaussian curvature* $K = \kappa_1\kappa_2$; the product of curvatures along two principal directions 1 and 2. To achieve large changes in $K$, such non-isometric deformations usually strive towards a coupled bending and in-plane stretching [20–22]. Excluded are composites with programmable stability effects, e.g. bi-stable snapping systems [23–25].

New applications for anisotropic self-shaping composites would be made possible by structures that, in contrast, allow $\Delta K > 0$. This could include for example initially flat structures self-shaping towards dome-like shells. One of the most suitable materials for self-shaping applications at medium and large scale is wood [26,27]. Wood is a material that can readily be applied as a self-shaping composite thanks to its innate capacity of anisotropic swelling and shrinkage and does not need any modification or complicated synthesis [28]. In addition, it is a common and low-cost construction material with unrivaled environmental virtues such as full renewability, low embodied energy and natural degradability. However, similar to the case of anisotropic shaping composites mentioned above, self-shaping bilayer plates made out of wood typically adopt a saddle-like configuration, and thus suffer from the above-mentioned restriction $\Delta K < 0$ [14,29].

In this study, we show a way to overcome the mentioned limitations by self-shaping gridshell systems (see electronic supplementary material, movie S1), focusing on orthogonal and interconnected narrow wood bilayer strips. Each wood bilayer strip will strive towards shaping to its natural curvature when the humidity is changed, and ideally, the gridshell system will realize a shape where $\Delta K > 0$. However, the system's behaviour, especially the exerted effect of rigid interconnections on the shaping, is yet unknown. In this context, we analyse the self-shaping behaviour of wood gridshell structures at medium scale by parametric numerical studies using the finite-element (FE) method. By investigating principal curvatures, natural curvatures, mean and Gaussian curvatures, we quantify and discuss the phase-space-dependent shaping behaviour and draw relevant conclusions for application to self-assembly architecture and construction processes. For such applications, gridshell systems offer a lightweight and material efficient method of covering large spaces with structurally performative curved forms [30–36].

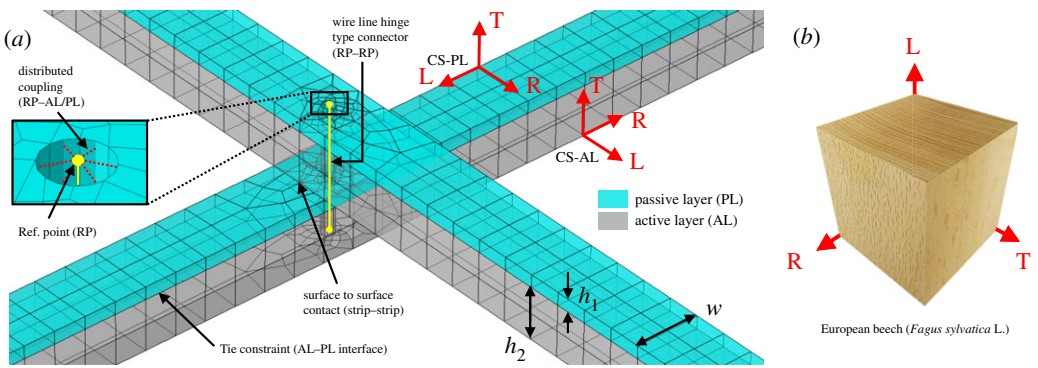

**Figure 2.** Gridshell FE model set-up. (*a*) Detail of FE model at strip intersection with parameters $h_1$, $h_2$ and $w$. Local orthotropic coordinate systems (CS) R, T and L assigned to passive and active layers (PL and AL) of strips. (*b*) Local anatomical directions R, T and L visualized on a macroscopic cube of European beech wood.

## 2. Methods

### 2.1. Gridshell design and parametric model framework

Self-shaping gridshells, half a metre wide and composed of 3×3 intersecting wood bilayer strips made out of European beech were investigated in a modelling framework. In a 3×3 grid, rigid-body and symmetric angle changes between the strips are prevented, and the system serves as an elementary representative of a rigidly interconnected larger grid. The gridshells were modelled to start in a flat state at an initially high wood moisture content (WMC) of 20%, and then self-shape while drying to 12% WMC. This roughly corresponds to conditioning the wood at 80% and 50% relative humidity climates at 20°C [37]. Both the thickness (*h*) and the width (*w*) of the strips were varied inside an interval of 5–25 and 5–45 mm (initial dimensions). The boundaries of *h* and *w* were chosen so as to represent application-based and feasible configurations at the medium scale for a proof of concept. In fact, wood strips with *h*, $w < 5$ mm are difficult to efficiently machine and $h > 25$ mm and $w > 45$ mm already represent building-scale dimensions. A single bilayer strip is composed of a passive layer (PL) with thickness $h_1$ and width $w$ connected (or glued) to an active layer (AL) of thickness $h_2$. The ratio $h_1 : h_2$ was constantly maintained to be 1:4, which for beech wood approximately corresponds to the ratio maximizing the natural curvature and minimizing elastic energy [19]. The gridshells were modelled using the FE method with the commercial software Abaqus 6.14. The model was set up by parametric Python scripting, which allowed for covering the above-described parameter space using an automatic computational framework described in electronic supplementary material, figure S2. The corresponding code files containing all necessary information on the used framework can be found in electronic supplementary material, document S4.

### 2.2. Model and boundary conditions

The moisture change from 20% to 12% WMC, inducing the shaping, was modelled by a steady-state change in field variable with automatic incrementation and with a nonlinear-geometry static analysis. The model was meshed with quadratic brick elements with reduced integration and $2 \times 2 \times 2$ integration points. The three local wood anatomical directions, radial (R), tangential (T) and longitudinal (L) were assigned to the PL and AL as shown in figure 2 so that the wood fibre direction L corresponded to the strip direction in the PL and to the perpendicular direction (in the grid-plane) in the AL. PL and AL were connected by tie constraints to form the strips. The interaction of the strips was modelled by a surface-to-surface contact (finite sliding, no tangential friction, normal separation enabled). Furthermore, the strips were tied together by a wire-line with hinge-connector properties (rotational degree of freedom about the grid out-of-plane axis), which was coupled to the inner surface of cylindrical holes of 2 mm in diameter through the middle of the strips (distributed coupling) through reference points. This set-up represents no-slip metallic screw-connections between the strips of the grid and was chosen to avoid further contact problems in the model. The grid is held by a vertical support and a pin at the extremities of a single mid-strip.

**Table 1.** Coefficients for calculation of moisture-dependent elastic engineering constants. The elastic engineering constants $P_i$ calculated as $P_i = b_0 + b_1\omega$ for European beech wood [38,39] and for $\omega$ in %. Young's and shear moduli $E_i$ and $G_i$ in units of MPa, Poisson ratios $\nu_i$ dimensionless.

| | $E_R$ | $E_T$ | $E_L$ | $G_{RT}$ | $G_{RL}$ | $G_{TL}$ | $\nu_{TR}$ | $\nu_{LR}$ | $\nu_{LT}$ |
|---|---|---|---|---|---|---|---|---|---|
| $b_0$ | 2566 | 885 | 17137 | 668 | 1482 | 1100 | 0.293 | 0.383 | 0.337 |
| $b_1$ | −59.7 | −23.4 | −282.4 | −15.2 | −15.3 | −17.8 | −0.001 | −0.009 | −0.009 |

## 2.3. Material model

The wood species European beech (*Fagus sylvatica* L.) was modelled as a linear hygro-elastic and orthotropic material with distinct material properties in directions R, T and L, and dependent on WMC, denoted $\omega$. The nine independent engineering constants of the fourth-order elastic stiffness tensor $\mathbf{C}^{el}$ used for the analysis are shown in table 1. The hygro-expansion and retraction coefficients $\alpha_i$ used in the analyses, also termed differential swelling and shrinkage coefficients, were chosen as $\alpha_R = 0.0019$, $\alpha_T = 0.0040$ and $\alpha_L = 0.0001$ in units of $\%^{-1}$ [26,40]. The parameters $\alpha_i$ represent elements in the diagonal hygro-expansion coefficient tensor $\boldsymbol{\alpha}$. The overall material behaviour is characterized by the free energy function

$$\Psi = \frac{1}{2}\,\boldsymbol{\varepsilon}^{el} : \mathbf{C}^{el} : \boldsymbol{\varepsilon}^{el}, \tag{2.1}$$

where the Cauchy stress tensor $\boldsymbol{\sigma}$ is derived from the material law as

$$\boldsymbol{\sigma} = \frac{\partial \Psi}{\partial \boldsymbol{\varepsilon}^{tot}} = \mathbf{C}^{el} : (\boldsymbol{\varepsilon}^{tot} - \boldsymbol{\varepsilon}^{\omega}) = \mathbf{C}^{el} : \boldsymbol{\varepsilon}^{el}. \tag{2.2}$$

Hereby, the total strain tensor calculates as $\boldsymbol{\varepsilon}^{tot} = \boldsymbol{\varepsilon}^{el} + \boldsymbol{\varepsilon}^{\omega}$ and the hygro-expansion strain tensor is $\boldsymbol{\varepsilon}^{\omega} = \boldsymbol{\alpha}\,d\omega$ for a moisture increment $d\omega$.

## 2.4. Output processing

Curvatures $\kappa_1$ and $\kappa_2$ along directions 1 and 2 (see coordinate system in figure 3) were calculated by automatic retrieving of coordinate lines on the deformed mid-strips by the inverse of radii of circles fitted to the coordinate lines. Respective natural curvatures $\kappa_0$ of the strips were calculated by two-dimensional plane stress FE models of single strips with set-up and boundary conditions matching the above description.

## 2.5. Physical demonstrators

Five experimental samples, serving as physical demonstrators at the extremities of the parameter space (illustrated in figure 3a by boxes A–E), were manufactured from European beech wood according to the exact boundary conditions described above for the computational model. The strip's PLs were glued to the ALs using a one-component polyurethane adhesive (1CPUR, HB S309 Purbond, Henkel & Cie. AG, Switzerland).

## 3. Results and discussion

Curvatures of the gridshell configurations after self-shaping by drying are shown in figure 3 along the initial and deformed states of the grids at the extremities of the phase-space (configurations A, C, D and E) for both simulated and experimental samples. The experimental samples displayed alongside the deformed model configurations (figure 3f) serve as qualitative comparison. In fact, the validity of the modelling approach used in this study has already been demonstrated in previous studies on hygromorphic beech wood bilayer strips [19,26]. Therefore, the following discussion will be based solely on the results obtained from the presented simulated study.

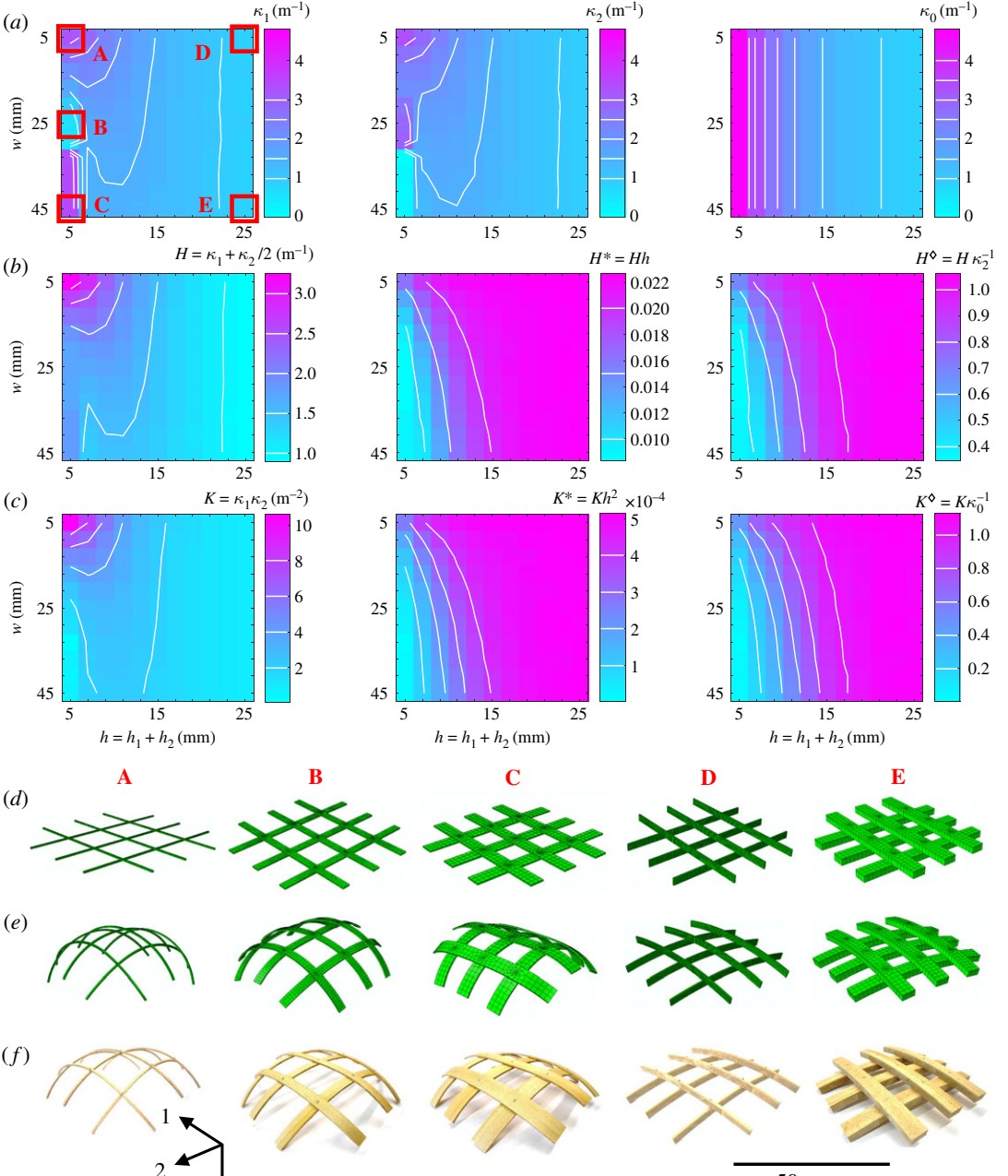

**Figure 3.** Phase space of gridshell curvatures after self-shaping in dependence of width $w$ and thickness $h$ for 121 (11×11) finite-element model solutions. (*a*) Curvatures along middle strips in directions 1 and 2 ($\kappa_1$ and $\kappa_2$) and natural curvatures $\kappa_0$. A, B, C, D, E denote configurations at extremities of analysed phase space. (*b*) Mean curvatures $H$, and mean dimensionless curvatures $H^*$ and $H^\diamond$. (*c*) Gaussian curvatures $K$, and dimensionless Gaussian curvatures $K^*$ and $K^\diamond$. (*d*) Configurations A, B, C, D, E in initial state (wet). (*e*) Configurations after self-shaping (dry). (*f*) Experimental configurations, 1 : 1 physical model rebuilds of the FE models using beech wood.

The graphs of the middle-strip gridshell curvatures $\kappa_1$ and $\kappa_2$ show that narrower strips and lower thicknesses result in increasingly curved gridshells. E.g. configuration A reaches $\kappa_1 = \kappa_2 = 3.2 \text{ m}^{-1}$, which corresponds to a radius of curvature of approximately 31 cm. A change in shaping mode can be observed for $h = 5$ mm at $w \approx 30$ mm. The gridshell curvature $\kappa_1$ is zero for $w < 30$ mm while $\kappa_2$ is high. This behaviour is inverted abruptly for $w > 30$ mm. The gridshells B and C illustrate this phenomenon. This behaviour can be explained by the fact that in either mode (B and C) the system's energetic cost of restraining the whole set of strips along a certain axis and allowing the other strips (along the other axis) to freely bend is lower than restraining both $\kappa_1$ and $\kappa_2$ to be $<\kappa_0$. The fact that

these modes apparently depend on $w$ for $h = 5$ mm but not for $h > 5$ mm can be explained by the strongly increasing contribution of $h$ on the residual elastic energy of shaping [19]. This implies a transition from $w$ to $h$ in dictating morphogenesis. In addition, the shaping of single strips is known to increase with a thickness ratio $h_1 : h_2$ close to the optimal ratio [19]. However, it can be observed here that the shaping of the gridshells is prevented by an increasing width because of rigid restraints at the screw joints of the strips. In fact, at lower values of $h$, thinner strips tend to bend in-plane and thus allow the rigid screw connections to spatially slip with respect to each other, on an imaginary shell plate surface. This allows for the system to realize a coupled bending and in-plane stretch for increased overall change in $K$. The in-plane bending of a border strip from configuration A is visualized in electronic supplementary material, figure S3 for both computational and physical models. The physical models qualitatively show similar shaping mechanism as the numerical models.

In the graph of the mean curvature $H$ (the arithmetic average of $\kappa_1$ and $\kappa_2$), the effect of strip width $w$ on shaping becomes apparent and it is revealed that wider strips increasingly reduce gridshell curvature for lower values of $h$. The plot of mean dimensionless curvature $H^* = Hh \approx \varepsilon_m$ reveals a dependence of average bending strain $\varepsilon_m$ on configuration. It should be noted here that the value $\kappa_0 h$ is constant for all $h$ and $w$ because of the compatibility $\kappa_0 = h^{-1}\varepsilon_0$ where $\varepsilon_0$ is the bending strain of an infinitely narrow strip. Therefore, the investigated gridshell configurations tend towards higher bending strains in proportion to $h$ as $h$ itself is increased, and a dependence on $w$ is observed for $h < 10$ mm. Alternatively, and instead of showing the amount of gridshell shaping in terms of bending strain alone, the dimensionless plot $H^\diamond = H\kappa_0^{-1} \approx \varepsilon_m \varepsilon_0^{-1}$ represents the shaping behaviour in terms of curvature realized in proportion to natural curvature of single strips, or in terms of average bending strain achieved in proportion to natural bending strain. It can be seen that an increasing thickness of the strips contributes towards the gridshell shaping to a higher extent where $H$ reaches up to approximately 105% of the value of $\kappa_0$. However, at around $h = 10$ mm, this behaviour is critically inverted and at around $h = 5$ mm, $H$ reaches approximately 40% of $\kappa_0$ only. We explain this behaviour by the fact that for thicker and wider strips, the bending moments, instead of resulting in in-plane deformations as for the narrow and thin strips, result in further out-of-plane deformations of the strips in a manner as to increase their curvature so that $\kappa_1$, $\kappa_2 > \kappa_0$. Therefore, configurations where $H^\diamond = H\kappa_0^{-1} > 1$ are not impossible. However, we note that by varying $w$, the proportional spacing of the strips in the grid with respect to $w$ is automatically varied too, and that this influence regarding configurations where $H^\diamond = H\kappa_0^{-1} > 1$ is not determined.

The plot of achieved double-curvature, i.e. the Gaussian curvature $K = \kappa_1 \kappa_2$ shows a very similar trend to that of the mean curvature $H$. However, it is apparent here that the configurations B and C effectively do not achieve double-curvature. In general, thicker and wider strips result in lower values of $K$. Similar to $H^*$, the graph of $K^* = Kh^2$ surprisingly shows that the double-curvature scaled over squared thickness is higher for thicker strips. Application-wise, e.g. for load-bearing self-shaped and double-curved gridshells structures, this suggests that an optimum of slenderness of strips compared to double-curvature benefits, i.e. bending moment-free members, may exist. Finally, in the same manner, the dimensionless double-curvature $K^\diamond = K\kappa_0^{-2}$ shows a matching trend with $H^\diamond$, leading to the conclusion that the qualitative behaviour of mean gridshell curvatures corresponds to that of the Gaussian curvatures in the case where the gridshells adopt a double-curvature (e.g. configurations A, D and E).

As was shown above, double-curvature can successfully be achieved by fine-tuning thickness and width of the strips, and therefore, self-shaping gridshells represent a promising application potential. Specifically, for timber gridshells in architecture, the double curvature is a decisive advantage that ideally enables bending-moment-free members under self-weight. In this context, the presented principle of self-shaping would be able to resolve some classical challenges inherent to timber gridshells. In fact, standard construction methods like pre-forming of the curved strips require intensive processes of forming and bending. Post-forming of a gridshell, a further method, can be achieved by a combination of lifting and elastically bending (active bending [41]) the structure into a fixed position. However, heavy edge constraints are required to confine the elastic bending forces in the strips [42,43] (see in figure 4). A type of self-forming process using gravity and mechanical locking of the individual layers has been previously shown at conceptual level but still requires additional machining and forcing of the structure into the curved shape [44]. In addition, conventional gridshell design is limited by the achievable cold-bending radius with respect to the lamella thickness and the wood quality [45,46]. This presents a major barrier for the application of a materially and sustainability effective construction system. Shaping a structure through the arrangement and material programming of its individual elements, like in the self-shaping process (figure 4), alludes to new types of simple, yet sophisticated deployable structures at medium to larger scale (0.1 to 10 m) and

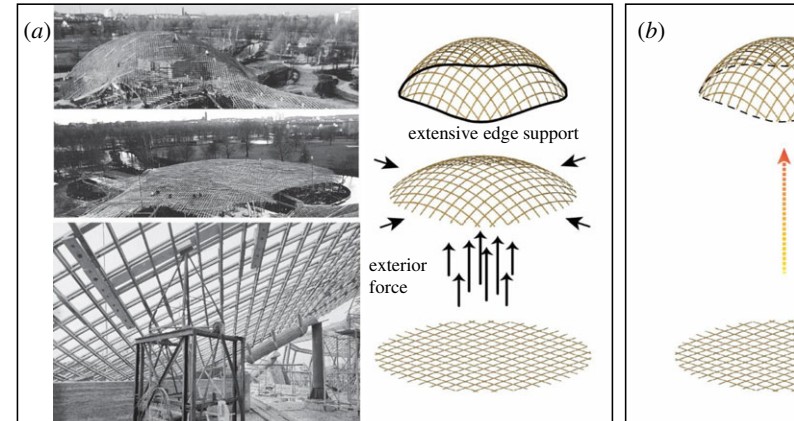
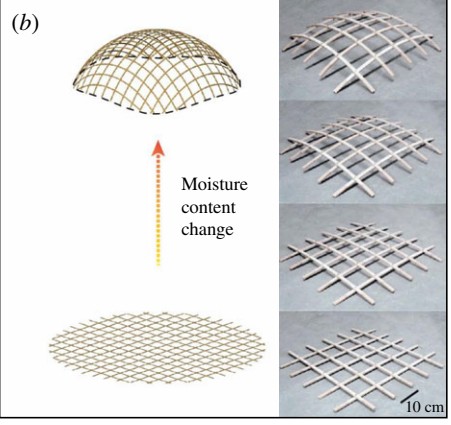

**Figure 4.** Gridshell shaping procedures. (*a*) Conventional procedure by shaping the gridshell via exterior forces for active elastic bending; Extensive edge support is needed in the assembled configuration (Photo credit: Institute for lightweight structures and conceptual design, University of Stuttgart). (*b*) Novel proposed procedure by hygromorphic self-shaping of the single strips (right side: Built 5×5 grid physical demonstrator); no exterior forces and lighter edge support.

addresses the mentioned problems restricting conventional gridshell design and assembly. Most importantly, self-shaping can deplete the need for tedious elastic shaping by exterior forces.

In order to realize such applications, some challenges remain to be addressed. In self-shaping gridshells, the timber strip members would be inherently composed of a large proportion of the strips with wood of grain direction being transverse to the strip direction. In the case of the investigated gridshell cross-sections above, for example, a 4/5 proportion is composed of wood with perpendicular grain direction. Therefore, large-scale load-bearing gridshells might be prone to increased buckling risks because of the normal forces in the members [47]. It could be of relevance, for example, how the *a priori* reduced critical buckling load depends on the grid spacing parameters affecting the buckling length. Therefore, the possible phase-space for large-scale applications of self-shaping timber gridshells might be reduced because of design restrictions with respect to stability. Nevertheless, such aspects highly depend on the material or wood species used [45,46], and European beech typically shows higher strength and stiffness than the common traditional softwoods used for timber gridshells [48]. Furthermore, the load-bearing behaviour is mainly dictated by spacing of strips and shape of the grid, which can be adapted to find suitable designs. And on top, further design and detailing possibilities of the actuating strips, e.g. adding passive layers that initially slide and then lock [44], could prevent limitations arising with respect to perpendicular grain or reversibility of shape.

Timber gridshells, thanks to their slenderness, can also be formed to adopt complex-curved shapes beyond uniform double curvature [49,50]. Such complex shapes are also conceivable with self-shaping. However, in this context, the programming of self-shaping gridshells by numerical analysis will need to be integrated in application-based form-finding procedures that are traditionally used for standard design of gridshell geometry under service loading conditions [51–56]. Especially, the minimizing of overall strain energy, here induced by the self-shaping, and with it the additional contributions of in-plane bending moments identified above, might be relevant for form-finding design [57]. Furthermore, and in contrast to the investigated configurations of this study, an even more effective achieving of a uniform $\Delta K > 0$ by self-shaping with respect to strain energy minimization might be achieved by an in-plane slip degree of freedom at the connections of the strips. While the double curvature of the gridshell depends on interplay of width and thickness of the single strips (as was shown in figure 3), a further and decisive parameter influencing global curvature is the allowed degree of freedom of deformation at the connections. This can be modelled by simply adapting the type of connector in figure 2. A scale-dependency of the shown results can be expected because of the rigidity of the connectors and its interplay with the scale-dependent mechanical behaviour of the material wood.

While the analysis in this study focused on simple symmetric gridshell configurations with rigid connections, the nowadays available design tools, including the computational framework provided by this study, enable a simple further tuning and variation of the strip geometry by efficient parametric analysis. Such tools of digital design, coupled to digital robotic fabrication, could be used to shape more complex shell forms at large and medium scale for future architectural applications.

Furthermore, the highly non-trivial but complete parametric space of a self-shaping gridshell would include the number and spacing for the strips, variation in layer thickness ratio of the strips, variation in initial and target moisture contents, and variation in wood species and local anisotropic material orientations in the layers. Nonetheless, the presented analysis with its implications for application may be readily applied to other self-morphing material systems and composites beyond wood, simply by adapting the boundary conditions and the material law.

# 4. Conclusion

Our results show that self-shaping wood gridshells enable positive changes in Gaussian curvature ($K$), which is impossible for solid plate geometries of single bilayer-composites of anisotropic materials. The shaping behaviour appears to be dictated by complex interactions between the strip's thickness and width. On the one hand, for thin and narrow strips, the large change in $K$ is enabled by the in-plane bending deformations of the strips of the rigid-jointed grid. On the other hand, for thicker and wider strips with lower changes in $K$, in-plane bending moments enforce the system's shaping towards values close to or slightly surpassing the natural curvature of single strips. The presented self-shaping would overcome many of the constraints of post-formed timber gridshells in the sense that complex lifting and forming is replaced with distributed autonomous actuation. Less structure is needed to constrain the shell after forming, tighter radii of curvature, and more complex lamella interactions would be made possible.

Data accessibility. All raw data and code supporting the results in this article can be found in electronic supplementary material, document S4.

Authors' contributions. Conceptualization, P.G.; methodology, P.G. and F.K.W.; formal analysis, P.G. and P.P.; writing–original draft preparation, P.G.; writing–review and editing, P.G., P.P., D.W., A.M., M.R. and F.K.W.; visualization, P.G. and D.W.; supervision, P.G., D.W., A.M., M.R. and F.K.W.; funding acquisition, M.R.

Competing interests. The authors declare no conflict of interest. The funders had no role in the design of the study; in the collection, analyses, or interpretation of data; in the writing of the manuscript, or in the decision to publish the results.

Funding. This research was funded by Innosuisse – Swiss Innovation Agency (grant no. 25114.2) and the APC was funded by Empa – Swiss Federal Laboratories for Materials Science and Technology. In addition, D.W. and A.M. acknowledge the partial support by the Deutsche Bundesstiftung Umwelt DBU (grant no. 34714/01) and the German Research Foundation DFG under Germany's Excellence Strategy (grant no. EXC 2120/1 – 390831618).

Acknowledgements. We sincerely thank Ingo Burgert for having provided the infrastructure and resources of the Wood Materials Science lab at ETH Zurich.

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
