## [Reviewer comments · Royal Society Open Science]

Review History

RSOS-192210.R0 (Original submission)

Review form: Reviewer 1

Is the manuscript scientifically sound in its present form?

Yes

Are the interpretations and conclusions justified by the results?

Yes

Is the language acceptable?

Yes

Do you have any ethical concerns with this paper?

No

Have you any concerns about statistical analyses in this paper?

No

Recommendation?

Major revision is needed (please make suggestions in comments)

Comments to the Author(s)

The provided manuscript inhibits an interesting approach of an innovative structural wood assembly and its self-shaping abilities to dome structures. The methods and results are explained and discussed in proper quality. The methods chosen are adequate, however, it is not fully clear whether empirical verification of the model has been conducted or not?

The discussion deals with the observed results from the model quite precisely but does not really state its originality. What is special about the findings? Would it really be applicable or what strengths and weaknesses does the model have or still need to overcome? Would it work with a thermally actuated bi-layers grid as well or is there something special of the chosen material? What is different from basic mechanics facts as e.g. second moment of area where height has a higher influence on bending than width? What are the authors' suggestions for further improvement of the model or for applying it to real constructions?

These are just minor complaints.

Based on the quality of scientific work provided, it is not comprehensible why the authors chose for an unconventional structure of the manuscript, as it is very hard to follow the context and interpret the conclusions. The current manuscript reads as the authors had written it in the usual order of 1. Introduction, 2. Material and Methods, 3. Results and Discussion 4. Conclusion and then re-organized it to the present structure, where the red thread and logical order got lost. I would hardly advise the authors to bring the manuscript into a structure where the scientific reader can easily follow the context. This is the major problem I have with this paper. Therefore, I suggest major revisions before publishing.

Further minor issues:

Page 4 line 23 to 39. Reads like material and methods but with missing details, which makes it hard to follow. Please put this into material and methods.

Page 4 line 30: Does the stated thickness and width correspond to 80 or 50% RH? What about the shrinkage of the material during the drying process? The shrinking coefficient was considered for h_2 in the radial direction for the bending? Was the reduction of h_2 due to transverse shrinking considered or neglected?

Page 4 line 38: What is the difference between the simulated and the experimental samples? Have samples been manufactured for an empirical verification? If yes please describe how they were produced, what moisture content before production. What glue used etc.

Page 7: 3. Application partly reads like the introduction. Consider combining.

Review form: Reviewer 2

Is the manuscript scientifically sound in its present form?

Yes

Are the interpretations and conclusions justified by the results?

Yes

Is the language acceptable?

Yes

Do you have any ethical concerns with this paper?

No

Have you any concerns about statistical analyses in this paper?

No

Recommendation?

Accept with minor revision (please list in comments)

Comments to the Author(s)

The submitted manuscript includes a computational analysis of self-shaping wood grid-shells and investigates effects of several geometrical parameters of the wooden strips on different curvatures of the 3-dimensionally deformed grid shell. This is a scientifically strong, very interesting and novel work and the manuscript is well structured. Methods, results and possible application are clearly described, illustrated and complemented by supplementary material describing implementation. I have only some minor comments and questions that hopefully can make the work even clearer.

One specific aim of the work was to study effects of "rigid interconnections" on the shaping. These connections obviously induce kinematic constraints in the structure and I wonder about corresponding reaction forces in the connections and relative rotations. The latter should be possible since the connection was modelled as hinged, i.e., strips can rotate relative to each other without relative displacement. I think this would also clarify what the term "rigid connection" implies in your study. I understand that the work focuses on the global curvature of the grid-shell, which is however a result of local constraints.

p4, line 57: "... shows similar magnitude of shaping ... as the numerical model." This is a qualitative statement and I wonder if a quantitative would be possible based on your physical experiments.

P 5, line 38: "... the presented analysis is deemed independent of scale ..." The influence of kinematic constraints on your structures might be important when thinking of different scales. I would appreciate more discussion about possible size-effects if a larger grid is considered?

P7, line 51: It could be good to emphasize that loading of structural elements of the grid-shell certainly depends on its shape, grid spacing and the global loading.

Python script for Abaqus modeling:

Some terms of the input parameters are given in German - it would be good to translate to English.

Decision letter (RSOS-192210.R0)

06-Apr-2020

Dear Mr Grönquist,

The editors assigned to your paper ("Computational analysis of hygromorphic self-shaping wood gridshell structures") have now received comments from reviewers. We would like you to revise your paper in accordance with the referee and Associate Editor suggestions which can be found below (not including confidential reports to the Editor). Please note this decision does not guarantee eventual acceptance.

Please submit a copy of your revised paper before 29-Apr-2020. Please note that the revision deadline will expire at 00.00am on this date. If we do not hear from you within this time then it will be assumed that the paper has been withdrawn. In exceptional circumstances, extensions may be possible if agreed with the Editorial Office in advance. We do not allow multiple rounds of revision so we urge you to make every effort to fully address all of the comments at this stage. If deemed necessary by the Editors, your manuscript will be sent back to one or more of the

original reviewers for assessment. If the original reviewers are not available, we may invite new reviewers.

- Data accessibility

If you wish to submit your supporting data or code to Dryad (<http://datadryad.org/>), or modify your current submission to dryad, please use the following link:
<http://datadryad.org/submit?journalID=RSOS&manu=RSOS-192210>

- Competing interests

- Authors' contributions

AB carried out the molecular lab work, participated in data analysis, carried out sequence alignments, participated in the design of the study and drafted the manuscript; CD carried out

the statistical analyses; EF collected field data; GH conceived of the study, designed the study, coordinated the study and helped draft the manuscript. All authors gave final approval for publication.

- Acknowledgements

- Funding statement

Kind regards,
Lianne Parkhouse
Royal Society Open Science
openscience@royalsociety.org

on behalf of Dr Maria Charalambides (Associate Editor) and R. Kerry Rowe (Subject Editor)
openscience@royalsociety.org

Reviewers' Comments to Author:

Reviewer: 1

Comments to the Author(s)

The provided manuscript inhibits an interesting approach of an innovative structural wood assembly and its self-shaping abilities to dome structures. The methods and results are explained and discussed in proper quality. The methods chosen are adequate, however, it is not fully clear whether empirical verification of the model has been conducted or not?

The discussion deals with the observed results from the model quite precisely but does not really state its originality. What is special about the findings? Would it really be applicable or what strengths and weaknesses does the model have or still need to overcome? Would it work with a thermally actuated bi-layers grid as well or is there something special of the chosen material? What is different from basic mechanics facts as e.g. second moment of area where height has a higher influence on bending than width? What are the authors' suggestions for further improvement of the model or for applying it to real constructions?

These are just minor complaints.

Based on the quality of scientific work provided, it is not comprehensible why the authors chose for an unconventional structure of the manuscript, as it is very hard to follow the context and interpret the conclusions. The current manuscript reads as the authors had written it in the usual order of 1. Introduction, 2. Material and Methods, 3. Results and Discussion 4. Conclusion and then re-organized it to the present structure, where the red thread and logical order got lost. I would hardly advise the authors to bring the manuscript into a structure where the scientific reader can easily follow the context. This is the major problem I have with this paper. Therefore, I suggest major revisions before publishing.

Further minor issues:

Page 4 line 23 to 39. Reads like material and methods but with missing details, which makes it hard to follow. Please put this into material and methods.

Page 4 line 30: Does the stated thickness and width correspond to 80 or 50% RH? What about the shrinkage of the material during the drying process? The shrinking coefficient was considered for h_2 in the radial direction for the bending? Was the reduction of h_2 due to transverse shrinking considered or neglected?

Page 4 line 38: What is the difference between the simulated and the experimental samples? Have samples been manufactured for an empirical verification? If yes please describe how they were produced, what moisture content before production. What glue used etc.

Page 7: 3. Application partly reads like the introduction. Consider combining.

Reviewer: 2

Comments to the Author(s)

The submitted manuscript includes a computational analysis of self-shaping wood grid-shells and investigates effects of several geometrical parameters of the wooden strips on different curvatures of the 3-dimensionally deformed grid shell. This is a scientifically strong, very interesting and novel work and the manuscript is well structured. Methods, results and possible application are clearly described, illustrated and complemented by supplementary material describing implementation. I have only some minor comments and questions that hopefully can make the work even clearer.

One specific aim of the work was to study effects of "rigid interconnections" on the shaping. These connections obviously induce kinematic constraints in the structure and I wonder about corresponding reaction forces in the connections and relative rotations. The latter should be possible since the connection was modelled as hinged, i.e., strips can rotate relative to each other without relative displacement. I think this would also clarify what the term "rigid connection" implies in your study. I understand that the work focuses on the global curvature of the grid-shell, which is however a result of local constraints.

p4, line 57: "... shows similar magnitude of shaping ... as the numerical model." This is a qualitative statement and I wonder if a quantitative would be possible based on your physical experiments.

P 5, line 38: "... the presented analysis is deemed independent of scale ..." The influence of kinematic constraints on your structures might be important when thinking of different scales. I would appreciate more discussion about possible size-effects if a larger grid is considered?

P7, line 51: It could be good to emphasize that loading of structural elements of the grid-shell certainly depends on its shape, grid spacing and the global loading.

Python script for Abaqus modeling:

Some terms of the input parameters are given in German – it would be good to translate to English.

Author's Response to Decision Letter for (RSOS-192210.R0)

See Appendix A.

RSOS-192210.R1 (Revision)

Review form: Reviewer 1

Is the manuscript scientifically sound in its present form?

Yes

Are the interpretations and conclusions justified by the results?

Yes

Is the language acceptable?

Yes

Do you have any ethical concerns with this paper?

No

Have you any concerns about statistical analyses in this paper?

No

Recommendation?

Accept as is

Comments to the Author(s)

All arisen queries were satisfactorily considered by the authors. I suggest accepting as is.

Review form: Reviewer 2

Is the manuscript scientifically sound in its present form?

Yes

Are the interpretations and conclusions justified by the results?

Yes

Is the language acceptable?

Yes

Do you have any ethical concerns with this paper?

No

Have you any concerns about statistical analyses in this paper?

No

Recommendation?

Accept as is

Comments to the Author(s)

The authors suitably addressed all my comments.

Decision letter (RSOS-192210.R1)

05-Jun-2020

Dear Mr Grönquist,

It is a pleasure to accept your manuscript entitled "Computational analysis of hygromorphic self-shaping wood gridshell structures" in its current form for publication in Royal Society Open Science. The comments of the reviewer(s) who reviewed your manuscript are included at the foot of this letter.

on behalf of Dr Maria Charalambides (Associate Editor) and R. Kerry Rowe (Subject Editor)
openscience@royalsociety.org

Reviewer comments to Author:
Reviewer: 2

Comments to the Author(s)
The authors suitably addressed all my comments.

Reviewer: 1

Comments to the Author(s)
All arisen queries were satisfactorily considered by the authors. I suggest accepting as is.

Appendix A

We sincerely thank the editors in charge, and especially the reviewers for their insightful comments. We would like to provide the following point-to-point reply to the points raised. Note that line and figure numbering used in the replies correspond to the new majorly revised version of the manuscript (not to the generated pages and line numbers by the submission system).

Reviewer: 1

Comments to the Author(s)

The provided manuscript inhibits an interesting approach of an innovative structural wood assembly and its self-shaping abilities to dome structures. The methods and results are explained and discussed in proper quality. The methods chosen are adequate, however, it is not fully clear whether empirical verification of the model has been conducted or not?

Empirical verification has been conducted in a qualitative manner. The experimental samples displayed in Fig. 3 serve for verification, whether the specific configuration adopts a double curvature or not. A quantitative verification of the gridshell configurations itself has not been conducted. However, we now have addressed the reviewer's question raised about the validity of the model used on page 5, lines 110-114, as this is a good point. In summary, the used model is indeed experimentally verified.

The discussion deals with the observed results from the model quite precisely but does not really state its originality. What is special about the findings? Would it really be applicable or what strengths and weaknesses does the model have or still need to overcome? Would it work with a thermally actuated bilayers grid as well or is there something special of the chosen material? What is different from basic mechanics facts as e.g. second moment of area where height has a higher influence on bending than width? What are the authors' suggestions for further improvement of the model or for applying it to real constructions?

These are all very relevant questions, that we tried to cover as good as possible.

To clarify: As mentioned above, the model of the single bilayer strip has been validated in previous studies. The novelty presented in this study is the self-shaping of grid shells using wood bilayers and the parametric analysis of the phase-space of shaping, i.e. whether double-curvature can be achieved or not in dependence on the parameters. Furthermore, the applicability in timber architecture is discussed. In the latter context, current limitations and further steps that need to be included in the modelling are discussed now throughout page 8 and 9, especially lines 219-230. A discussion about the used modelling technique itself, namely the Finite Element Method (FEM), is deemed out of context here.

The chosen material is wood and the mechanical behavior was modelled accordingly: Orthotropy and moisture-dependent entries of the compliance tensor, see Table 1 on page 4. The specialty of wood is its anisotropic swelling and shrinking behavior as a function of moisture content and the possibility to transform this dimensional change into shape change at large scale due to its mechanical stability. It is straightforward to adapt this principle, e.g. to thermally actuated bilayer grids made out of metals if desired. We now added this aspect in the manuscript with the sentence on page 9 lines 227-230.

The question about height vs width influence: This is one of the main questions addressed throughout the whole manuscript, which goes well beyond the respective influence on the magnitude of curvature. To

summarize the findings in the manuscript in this respect: Height influences magnitude of bending, whereas width influences whether double-curvature or not can be achieved (as stated on page 2, lines 2-21). The main result of our study, Fig. 3, page 7, showcases the interplay between both in the gridshell system. We already noted that this interplay is also highly dependent on the type of connection between the strips.

These are just minor complaints.

Based on the quality of scientific work provided, it is not comprehensible why the authors chose for an unconventional structure of the manuscript, as it is very hard to follow the context and interpret the conclusions. The current manuscript reads as the authors had written it in the usual order of 1. Introduction, 2. Material and Methods, 3. Results and Discussion 4. Conclusion and then re-organized it to the present structure, where the red thread and logical order got lost. I would hardly advise the authors to bring the manuscript into a structure where the scientific reader can easily follow the context. This is the major problem I have with this paper. Therefore, I suggest major revisions before publishing.

We appreciate the suggestion for an improved structure. Based on the reviewer's comments, we changed the structure of the manuscript to read "1. Introduction, 2. Materials 3. Results and Discussion 4. Conclusion", for a better clarity.

Further minor issues:

Page 4 line 23 to 39. Reads like material and methods but with missing details, which makes it hard to follow. Please put this into material and methods.

This section was now moved to the section "2. Materials" as suggested (to page 3, lines 47-65). Furthermore, for even better clarity in this section, we have added subsections (a)-(e).

Page 4 line 30: Does the stated thickness and width correspond to 80 or 50% RH? What about the shrinkage of the material during the drying process? The shrinking coefficient was considered for h_2 in the radial direction for the bending? Was the reduction of h_2 due to transverse shrinking considered or neglected?

The stated thickness and width correspond to initial conditions at 80%RH (this was described in the Materials section), but for better clarity, we have now added this additionally in parenthesis on Page 3 line 54. The shrinkage in all layers and in every space direction is indeed considered, as is described already on page 4 line 88, and by the coordinate systems of Fig. 2.

Page 4 line 38: What is the difference between the simulated and the experimental samples? Have samples been manufactured for an empirical verification? If yes please describe how they were produced, what moisture content before production. What glue used etc.

See reply above (first comment of reviewer 1): The aim of the study was a qualitative empirical verification of whether a double-curvature could be achieved or not. The production of samples is now newly described in subsection (e) of the section Materials, on page 5. The experimental sample production straightforwardly follows exactly the FE-model setup (and vice-versa), with exact same geometry and boundary conditions (moisture content etc.).

Page 7: 3. Application partly reads like the introduction. Consider combining.

We thank the reviewer for this suggestion, but prefer not to combine both sections in order to differentiate the motivation problematic (the limitation of self-shaping composites to narrow strips and our approach of by-passing a plate by a grid) with the final outcome and possibilities our study demonstrated and offers for application (successfully achieving double-curved gridshells in timber architecture).

Reviewer: 2

Comments to the Author(s)

The submitted manuscript includes a computational analysis of self-shaping wood grid-shells and investigates effects of several geometrical parameters of the wooden strips on different curvatures of the 3-dimensionally deformed grid shell. This is a scientifically strong, very interesting and novel work and the manuscript is well structured. Methods, results and possible application are clearly described, illustrated and complemented by supplementary material describing implementation. I have only some minor comments and questions that hopefully can make the work even clearer.

We sincerely thank reviewer 2 for the praise. However, we still changed the structure of our manuscript based on the comments above of reviewer 1.

One specific aim of the work was to study effects of "rigid interconnections" on the shaping. These connections obviously induce kinematic constraints in the structure and I wonder about corresponding reaction forces in the connections and relative rotations. The latter should be possible since the connection was modelled as hinged, i.e., strips can rotate relative to each other without relative displacement. I think this would also clarify what the term "rigid connection" implies in your study. I understand that the work focuses on the global curvature of the grid-shell, which is however a result of local constraints.

This is an interesting point raised by the reviewer. Indeed, reaction forces appear on the connections, however, as is correctly stated, these only allow a rotational degree of freedom. Interestingly, the rotation of the hinges is of minor concern considering the global gridshell curvature. The latter would be much more influenced by an in-plane slippage at the connections, which is however modelled as totally hindered. In fact, the kinematic of the grid system prevents any rotation of the strips at the hinges (this can be seen in supplementary figure Fig. S3), and the only way to achieve large in-plane stretches (i.e. to extensively change Gaussian curvature, as mentioned in the Introduction section on page 2) is when the strips perform in-plane bending (notice how the strips still maintain a 90° angle at the connections at all times). Page 3 lines 47-50 describes this with the motivation of having chosen a 3x3 grid. (However, in a 2x2 grid, rotation at the hinges are possible!). The fact that the gridshell curvature is a result of local constraints, as the reviewer correctly states, was indeed not mentioned clearly enough. A sentence stating this was now added on page 9 lines 213-216.

p4, line 57: "... shows similar magnitude of shaping ... as the numerical model." This is a qualitative statement and I wonder if a quantitative would be possible based on your physical experiments.

A quantitative comparison is unfortunately not possible. We therefore corrected the mentioned sentence on page 5 lines 133-134.

P 5, line 38: "... the presented analysis is deemed independent of scale ..." The influence of kinematic constraints on your structures might be important when thinking of different scales. I would appreciate more discussion about possible size-effects if a larger grid is considered?

This is a good point. We have now added more discussion considering the size-effect of constraint versus the problematic of size effect of wood mechanical behavior on Page 9 lines 216-218, which we estimate is the main effect affecting results if the scale is changed (mechanics of connection does not change with size but mechanics of strips does change, and their interplay is of relevance).

P7, line 51: It could be good to emphasize that loading of structural elements of the grid-shell certainly depends on its shape, grid spacing and the global loading.

This is worthy of mentioning. We have included this as a new sentence on Page 8 lines 198-200.

Python script for Abaqus modeling:

Some terms of the input parameters are given in German – it would be good to translate to English.

We thank the reviewer for noticing this. All German terms have now been changed to English.